# Earthquake nucleation in the lower crust by local stress amplification

L.R. Campbell [1✉], L. Menegon[1,2], Å. Fagereng [3] & G. Pennacchioni [4]

Deep intracontinental earthquakes are poorly understood, despite their potential to cause significant destruction. Although lower crustal strength is currently a topic of debate, dry lower continental crust may be strong under high-grade conditions. Such strength could enable earthquake slip at high differential stress within a predominantly viscous regime, but requires further documentation in nature. Here, we analyse geological observations of seismic structures in exhumed lower crustal rocks. A granulite facies shear zone network dissects an anorthosite intrusion in Lofoten, northern Norway, and separates relatively undeformed, microcracked blocks of anorthosite. In these blocks, pristine pseudotachylytes decorate fault sets that link adjacent or intersecting shear zones. These fossil seismogenic faults are rarely >15 m in length, yet record single-event displacements of tens of centimetres, a slip/length ratio that implies >1 GPa stress drops. These pseudotachylytes represent direct identification of earthquake nucleation as a transient consequence of ongoing, localised aseismic creep.

¹ School of Geography, Earth and Environmental Sciences, Plymouth University, Plymouth PL4 8AA, UK. ² The Njord Centre, Department of Geosciences, University of Oslo, P.O. Box 1048 Blindern, Norway. ³ School of Earth and Ocean Sciences, Cardiff University, Cardiff CF10 3AT, UK. ⁴ Department of Geosciences, University of Padova, 35131 Padova, Italy. ✉email: lucy.campbell@plymouth.ac.uk

Earthquake mechanics are mostly studied in the context of the brittle upper crust, where earthquakes predominately occur[1]. However, earthquakes also nucleate in the continental lower crust in mechanically strong lithologies[2–4]. Deep continental earthquakes tend to nucleate along intraplate faults, or faults cutting thick, cold cratons[5–8]. Earthquakes in continental interiors have resulted in significantly higher casualties than earthquakes at plate boundaries[9]. Thus, a thorough understanding of the earthquake cycle in intracontinental settings is essential, and requires knowledge of the mechanical behaviour and seismogenic potential of the lower crust.

The occurrence of pseudotachylytes (solidified frictional melts) formed at lower crustal conditions has been taken as geological evidence of both high mechanical strength and the occurrence of seismic rupture below the typical seismogenic zone[10–14]. Seismological observations of lower crustal earthquakes, e.g. in East Africa[2] and in the India-Tibet collision zone[3], compare favourably with geological studies suggesting a dry, metastable, strong and seismic lower crust[15,16].

Earthquakes in dry (e.g. <0.1 wt. % $H_2O$[17]) lower crustal rocks at depths ≥25–30 km require either transiently high differential stresses or local weakening mechanisms (e.g. high pore fluid pressure[5]). One explanation for transient high stresses is the downward propagation of an earthquake rupture from the shallower seismogenic zone[4,18]. These stress spikes account for transient post-mainshock deformation including rapid postseismic strain rates[19] and clusters of aftershocks beneath the mainshock rupture area[4,20,21]. Whilst large continental earthquakes that nucleate in the upper crust and propagate downwards to depths of 20–25 km are not uncommon[7,22–24], the mechanisms of earthquakes that nucleate within the lower crust are still intensely debated. Proposed mechanisms include thermal-runaway plastic instabilities[25], dehydration reactions leading to increased fluid pressure[26] and/or local stress redistributions[27], or eclogitisation reactions[28]. These examples, however, require syn-deformational reactions that may not be occurring in all locations hosting local, lower crustal seismicity. Here we suggest a mechanism where earthquakes nucleate within dry and strong lower crustal rocks without the need of syn-deformational reactions or seismic loading from shallower crustal levels, but rather as a direct consequence of loading of low strain domains during deformation along a network of intersecting ductile shear zones. This is an important advance in our understanding, because this mechanism can explain lower crustal seismicity in regions without shallow seismicity or evidence for fluids, such as deep earthquakes observed in the northern Central Alpine foreland[29].

We characterise an exhumed network of highly localised shear zones recording viscous deformation at lower crustal conditions. We describe the geometry of pseudotachylyte veins that cut between shear zones of varying orientations, outline the evidence that these markers of seismicity were coeval with viscous creep of the shear zones at lower crustal conditions, and use measurements of fault length and displacement to calculate the moment magnitudes and static stress drops of these seismic events. We interpret this mechanism of seismicity to be a mechanical response to strain incompatibility across the shear zone network during localised viscous shear, with strong blocks undergoing seismic failure at points of local stress amplification. This is the first evidence for in-situ, high stress drop earthquake nucleation in the lower crust driven predominantly by the geometry of a shear zone network, as a consequence of differential creep rates and high viscosity contrasts.

## Results

### The Nusfjord shear zone network.
The Lofoten-Vesterålen islands of Norway expose a 1.9–1.7-Ga Anorthosite-Mangerite-Charnockite-Granite suite that extensively preserves an anhydrous granulite facies assemblages[30]. In the SE of Flakstadøy (Supplementary Fig. 1) the Nusfjord coarse-grained anorthosite is cut by mylonitic shear zones that were active at 650–750 °C and 0.7–0.8 GPa[31].

The mylonitic shear zones, concentrated within an E-W striking high strain zone of ~1 km apparent thickness, occur in three main sets (Sets 1–3) that match the orientations of regional tectonic lineaments (Fig. 1). This high strain zone comprises mylonitic shear zones ranging from numerous narrow (typically 5–20 cm thick) structures to less frequent wider structures consisting of multiple shear zone strands (Figs. 1 and 2). Most Set 1 shear zones (average foliation dip/dip direction of 54°/156° and average lineation plunge/azimuth of 48°/184°) include mylonitised pseudotachylytes (type-1 pseudotachylytes) and have accumulated appreciable oblique normal displacement (Fig. 2). Set 2 shear zones strike NW-SE (average foliation 82°/050°, and lineation 18°/328°), contain fewer type-1 pseudotachylytes and show sinistral strike-slip movement. Set 3 shear zones are generally minor structures with strike varying from N-S to NE-SW (average foliation orientation of 88°/278°, Fig. 1).

The shear zones exploited precursor dykes and type-1 pseudotachylyte-bearing faults (Fig. 2a). Strain localisation into shear zones rather than the surrounding anorthosite was promoted by several mechanisms including: the reduced grain size (10–30 μm in the pseudotachylyte compared to ~10 cm in the surrounding anorthosite), the increased phase mixing in the pseudotachylytes, and the increased water content in the pseudotachylytes (0.4 wt.% vs 0.05 wt.% in the anorthosite[31]). All these mechanisms are inferred to promote grain size sensitive creep at lower stresses than required to deform the anorthosite[31]. In addition, there is a mechanical anisotropy introduced by the tabular geometry of these precursors. The coarse-grained anorthosite between the shear zones is microfractured (Supplementary Fig. 2) but does not show evidence of dislocation creep in the plagioclase or pyroxenes. Deformation microstructures indicative of local (and limited) dislocation glide occasionally occur in plagioclase and pyroxenes in the form of undulatory extinction and lattice distortion (Supplementary Fig. 2). In general, however, the anorthosite blocks show no evidence of internal high strain deformation. The high strain zone therefore consists of a network of variously oriented, intersecting, narrow shear zones separating blocks of barely deformed anorthosite (Fig. 1b, c). Coeval viscous creep on all shear zone orientations is indicated by mutually offsetting shear zones and convergent stretching lineations at intersection zones.

### Pseudotachylytes and shear zones.
Whilst type-1 pseudotachylytes are commonly mylonitised, type-2 pseudotachylytes are dominantly pristine veins, undeformed and unaltered from their origin as crystallised melts, inferred to be coseismic. Type-2 pseudotachylytes occur along small-displacement faults that dissect anorthosite blocks bounded by either subparallel or intersecting shear zones (Fig. 3). These shear zone-confined blocks are observed at length scales ranging from 1 m to 15 m (Fig. 3, Supplementary Fig. 3) and typically occur between Set 1 shear zones (Fig. 3a) or Set 1—Set 2 shear zone intersections (Fig. 3b). The confined type-2 pseudotachylyte-bearing faults neither offset, nor are they offset by, the bounding shear zones.

Figure 3a shows a type-2 pseudotachylyte-bearing fault network between two Set 1 shear zones, spaced ~10 m apart. The bounding Set 1 shear zones dip to the SE and show transtensional kinematics. Type-2 pseudotachylytes are locally dragged into the southern Set 1 mylonite (Fig. 3c), but outside the centimetre-thick dragging zone the original pull-apart geometry and pristine microstructures of type-2 pseudotachylytes are well-preserved (Fig. 3c, d). The

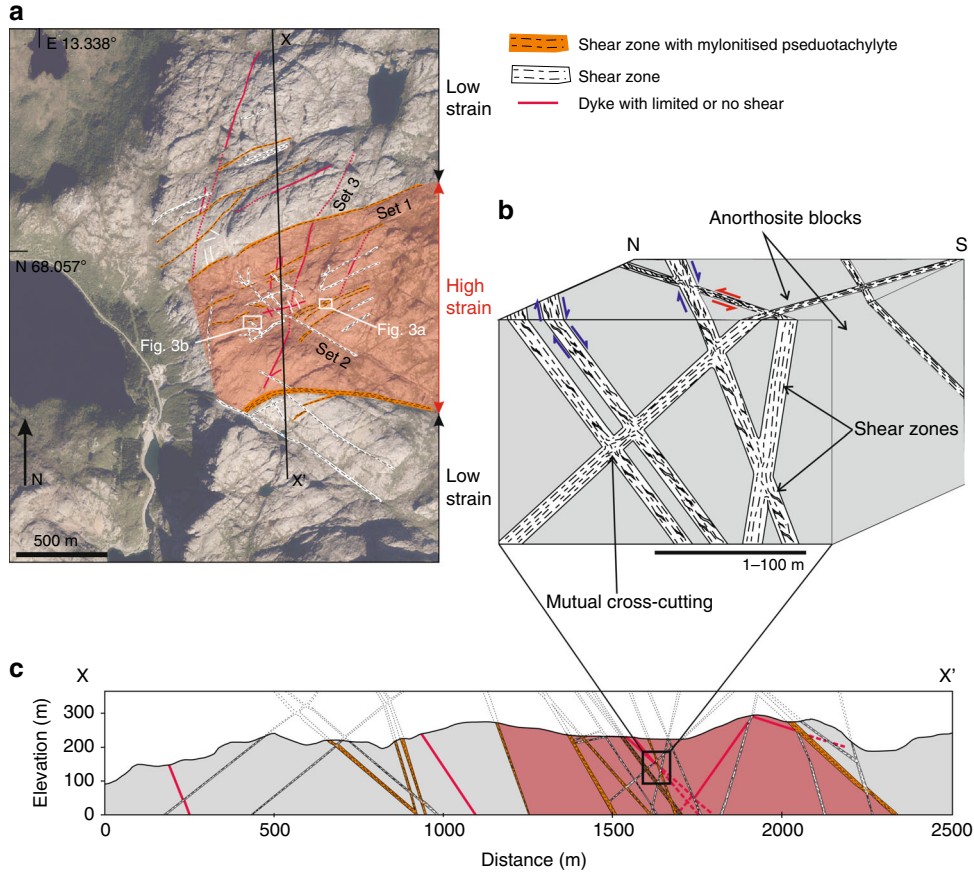

**Fig. 1 Structure of the Nusfjord region. a** Map of Nusfjord anorthosite showing locations of shear zones and large dykes. Red shading indicates the extent of the high strain zone containing frequent cross-cutting shear zones. Basemap © OpenStreetMap contributors (openstreetmap.org.uk/copyright). **b** Block diagram showing three-dimensional structure of interlocking shear zones in various orientations. **c** Cross section taken from line X–X′ shown on map in 1a. The thickness of shear zones in the map is not representative of the actual thickness in field.

mutually intersecting type-2 pseudotachylyte-bearing faults show a combination of dextral and sinistral separations on the outcrop surface. The dextral faults typically dip moderately NE to SE and show oblique normal-dextral kinematics; sinistral faults dip steeply towards north or south and are dominantly strike-slip (Fig. 3a). A similar structural geometry as shown in Fig. 3a continues to the NW and SE along-strike extension of the mapped area for about 100 and 200 m respectively (e.g. Supplementary Fig. 3a), though with some variations related to local segmentation and branching of the bounding shear zones.

In Fig. 3b, type-2 pseudotachylytes are confined between intersecting Set 1 mylonitised type-1 pseudotachylyte (with transtensional kinematics) and Set 2 sheared pegmatite (exhibiting left-lateral strike-slip). Type-2 pseudotachylytes are concentrated close to the shear zones' intersection. Within 1 m of the intersection, pseudotachylytes both cut across and are transposed along the shear zone foliation (Fig. 3e), whereas, at a greater distance, they extend from the Set 1 towards the Set 2 shear zone (Fig. 3b). These latter type-2 pseudotachylytes are associated with small faults typically with sinistral component of offset and variable orientation (dipping moderately SE and steeply NW and E (Fig. 3b).

In the examples of Fig. 3a, b, type-2 pseudotachylyte veins mostly preserve the pristine macroscopic geometry (e.g. en-echelon arrangement, pull apart jogs, chilled margins and equant clasts, Fig. 3f) and, especially in the vein core, the microscopic microlitic/spherulitic texture (Fig. 3d). However, in one case, microstructural analysis reveals that the pseudotachylyte vein

margins localised solid-state shearing over a width of 1 mm (Fig. 3d). This very discrete shearing is not easily observed in the field and does not account for any significant displacement. Other sampled type-2 pseudotachylytes do not show any viscous overprint (Supplementary Fig. 2a).

**Evidence for earthquake nucleation within the lower crust.** The confinement of these faults within relatively intact, shear zone-bounded blocks, with dragging of pseudotachylytes into the shear zones, implies that seismic ruptures were coeval with viscous shear. Mylonitisation along the bounding shear zones occurred at lower crustal conditions of 650–750 °C and 0.7–0.8 GPa, based on amphibole-plagioclase geothermobarometry and thermodynamic modelling of the mylonitised pseudotachylyte assemblages[31]. These conditions can thus be assumed also for the generation of the type-2 pseudotachylytes, supported by the stability of the granulite facies mineral assemblage (plagioclase+clinopyroxene+hornblende+orthopyroxene +garnet+biotite±quartz±K-feldspar) found both in the host rock damage zone, within the sheared margins of pseudotachylyte, and crystallised within the vein itself. Therefore, type-2 pseudotachylyte-bearing faults represent earthquakes nucleated under lower crustal conditions.

The concentration of type-2 pseudotachylytes near shear zone intersections reveals that earthquake slip was controlled by the interaction and geometry of shear zones. Concurrent slip of shear zones, delimiting polyhedral blocks of pristine anorthosite, forced low strain blocks to deform internally[32].

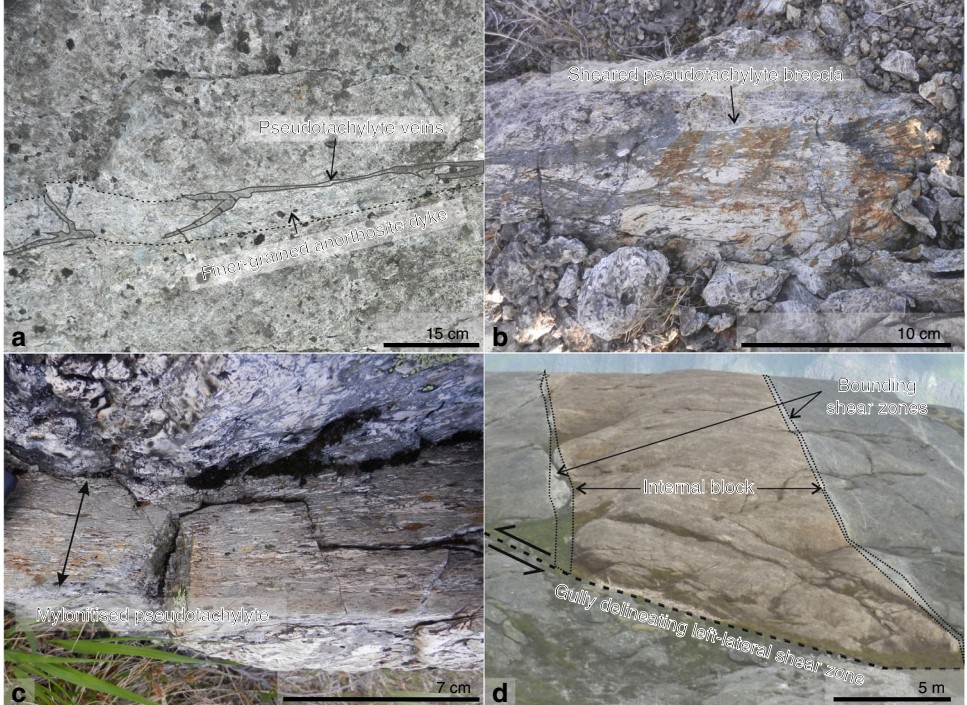

**Fig. 2 Shear zones and sheared pseudotachylytes in the Nusfjord anorthosite. a** Pseudotachylyte localised along the margin of a relatively fine-grained anorthosite dyke. A viscous overprint in the pseudotachylyte can be seen in the apparently sheared geometry of the injection veins [68.0563°N 13.3648°E]; **b** pseudotachylyte fault breccia with viscous overprint shown by alignment of deformed, elongate clasts parallel to the shear zone foliation [68.0572°N 13.3687°E]; **c** well-foliated mylonite consisting of significant proportions of pseudotachylyte—interpreted to be a high strain equivalent to the breccia in b. Outside the sheared pseudotachylyte, the anorthosite is undeformed [68.0572 °N 13.3687 °E]; **d** view looking west of an internal anorthosite block (with orange overlay) between two bounding shear zones emphasised in white and detailed in Fig. 3a [68.0557°N 13.3746°E].

**Earthquake source parameters**. Type-2 pseudotachylytes were formed by earthquake ruptures <~15 m in apparent length. Pseudotachylyte pull-aparts (Fig. 3f) record single-event displacements in the range of 1 to 26 cm, with the highest displacements observed along longer faults (Fig. 4a). The displacement/length ratio of these single-slip faults ranges between $10^{-3}$ and $10^{-1}$, well exceeding the typical ratios of $10^{-7}$-$10^{-4}$ seen in kilometre-scale earthquake ruptures (Fig. 4a)[22,33]. This may indicate that, due to interaction with the bounding shear zones, ruptures terminated prematurely. Assuming a circular rupture area, the type-2 pseudotachylytes record moment magnitudes ($M_w$) ranging from 0.2 to 1.8 and their slip/length ratios imply static stress drops between 0.1 and 4.2 GPa (Fig. 4b). This assumption of a circular rupture may underestimate the true rupture area. Although we have no observation of the vertical dimension of the pseudotachylyte-bearing faults, it is unlikely that their aspect ratio is >10, because the maximum vertical extent of anorthosite blocks with widths ~10 m is neither observed nor projected to extend much over 100 m (Fig. 1c). Therefore, an elliptical fault with a long axis ten times the measured fault length provides an upper bound for rupture area and moment magnitude ($M_w$ 0.8–2.6), and a lower bound for stress drop (0.06–2.5 GPa) (Fig. 4b, Supplementary Fig. 5).

These minimum stress drops calculated from type-2 pseudotachylytes are still high when compared to seismological records (Fig. 4b) of upper crustal-[34,35] and intracontinental lower crustal-seismicity[36,37]. These stress drops are also generally higher than other values calculated from pseudotachylyte-bearing faults[38], including those recording continental lower crustal and mantle seismicity[11,39], although are on the same order as those derived from pseudotachylytes at depths of 7–10 km[40] and in lawsonite-eclogite facies peridotites[41]. We note, however, that the calculated stress drops may be elevated because the rupture area is limited to

the size of the block, analogous to large stress drops seen in some laboratory shear experiments[42]. The large stress drops imply that the failure shear strength of the intact anorthosite must be >1 GPa, consistent with both the high strength of anorthite reported from experimental studies[43] and the stresses required for failure of intact anorthosite and subsequent frictional sliding at high lower crustal confining pressures. High viscous strength in the anorthosite blocks would enable seismic failure through elastic energy accumulation, because the dry, coarse-grained plagioclase could not flow viscously—even at geological strain rates—without a reduction in grain size, changed mineralogy, or fluid influx[43,44].

## Discussion

Our results imply that lower crustal earthquake nucleation may result from localised viscous creep along shear zone networks within dry granulitic lower crust. In the Nusfjord anorthosite, local high differential stresses are inferred to have arisen from the interaction of localised viscous shear zones at lower crustal levels, where a high viscosity block experienced stress amplification imposed by flow of the surrounding, weaker material (cf. refs. [45–48]).

We argue that strain incompatibility across the deforming system was accommodated by transient seismic failure along new faults nucleating at sites of stress amplification within the strong anorthosite blocks. Over long timescales, the effect of episodic seismic activity was to approximate strain compatibility across the shear zones, at least enough to facilitate ongoing viscous deformation. A similar model was hypothesised to explain the cyclic generation of pseudotachylytes in the lower crustal rocks of the Musgrave Ranges[14]. Here we provide the first evidence for

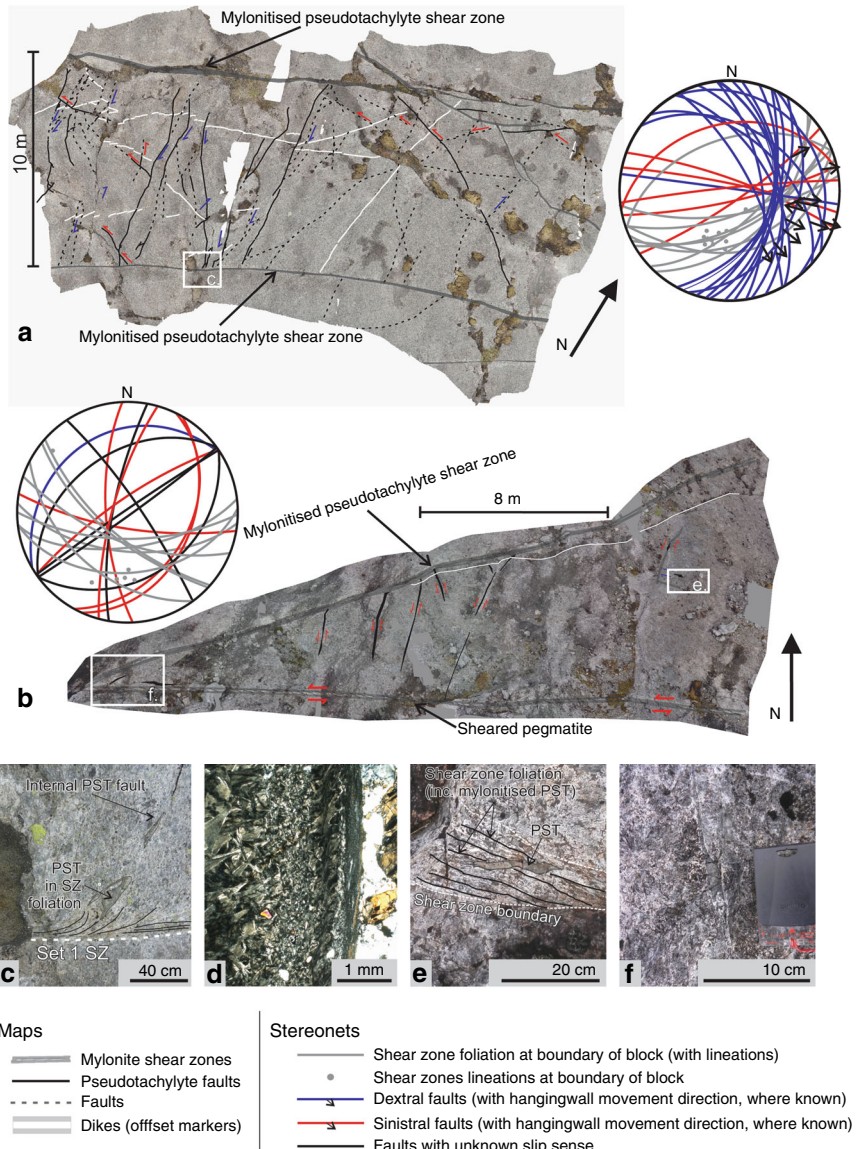

**Fig. 3 Type-2 pseudotachylyte faults within shear zone-bounded blocks. a** map of pseudotachylyte fault network developed between two SE-dipping Set 1 shear zones [68.0557 °N 13.3744°E] and stereonet of fault and shear zone orientations for the region (separated versions of the photo-map and sketch maps for parts a and b are available in Supplementary Fig. 4); **b** map of pseudotachylyte fault network developed between a SE-dipping Set 1 shear zone and a SW-dipping Set 2 shear zone [68.0552°N 13.3678°E] and stereonet of shear zone and pseudotachylyte orientations; **c** southern boundary shear zone of system in 3a shows pseudotachylyte faults dragged into shear zone foliation; **d** micrograph of type-2 pseudotachylyte vein showing transition from radiating microlites to fine-grained and viscously sheared margin of pseudotachylyte vein (cross-polarised light); **e** detail of the intersection between the two shear zones, including scattered pseudotachylytes which cross from the undeformed internal region into the shear zones and are partially transposed along the foliation; **f** sinistral stepover jog developed in type-2 pseudotachylyte.

such in-situ seismic lower crustal faulting based on detailed field maps of the Nusfjord ridge. Earthquake ruptures may be encouraged both by the interaction of differently oriented shear zones and by their differential creep rates. In this interpretation, seismic faulting took place as punctuated failure episodes constrained to individual internal blocks (Fig. 1a, c). Within each block, stresses increased as viscous creep on the bounding shear zones progressed alongside a continued absence of deformation in the internal block[10,45] (Fig. 5a). The magnitude of stress amplification would increase with increasing volumetric block to shear zone ratio, and with an increasing viscosity contrast between the blocks and the bounding shear zones[45,46]. Spatial heterogeneity of stress amplification was likely controlled by the geometry of the bounding shear zones and the internal block[32]. Progressively, continuing deformation along bounding shear

zones would have increased the geometrical strain incompatibility, and, in the absence of viscous deformation in the anorthosite, increasing elastic strains within the blocks would have locally increased shear stresses towards the anorthosite failure strength. Seismic rupture released the amplified stress via the coseismic stress drop. In this way, cycles of elastic stress accumulation and release occurred locally within each block in response to displacements on the bounding shear zones (Fig. 5a), but cumulative seismic failure across several blocks could (over some unknown time-scale) facilitate ongoing creep of the entire kilometre-wide high strain zone (Fig. 5b).

The absolute magnitudes of the proposed stress amplifications are difficult to estimate from comparisons of existing work due to differences in model geometry, rheology, deformation mechanisms, strains, and strain rates, and would benefit from further

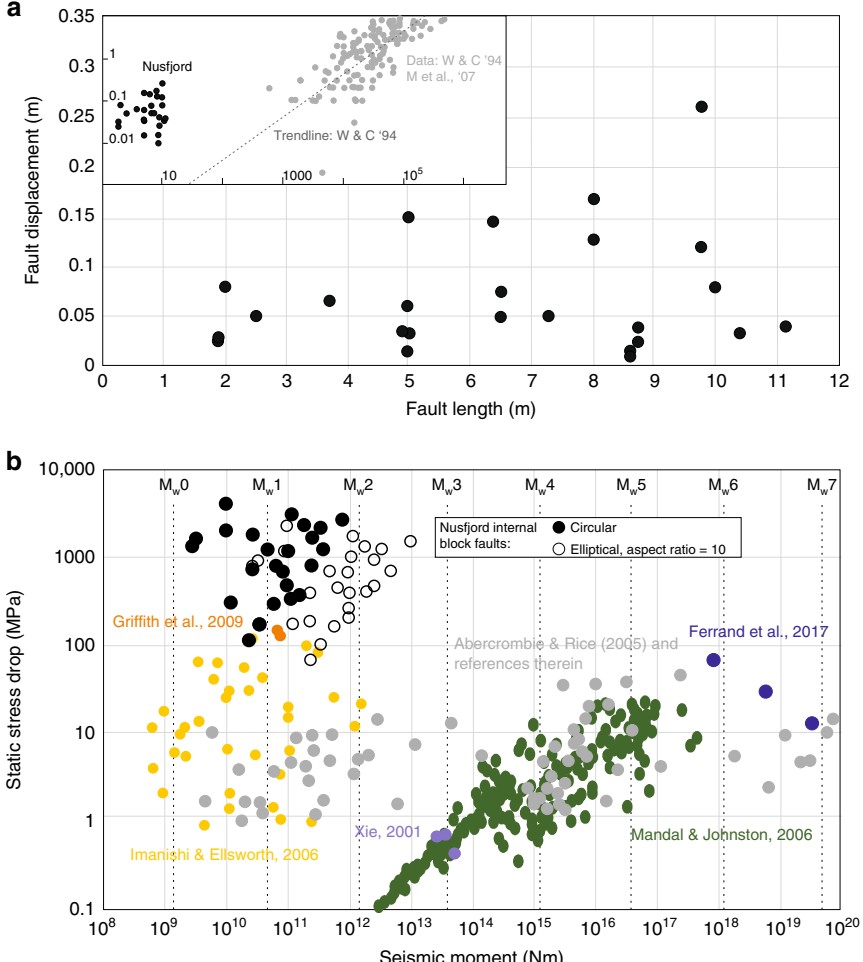

**Fig. 4 Seismic source parameters of type-2 pseudotachylyte. a** length vs displacement graph, inset compares Nusfjord type-2 pseudotachylytes with published data on rupture length versus displacement[22,33]; **b** seismic moment vs static stress drop with equivalent moment magnitudes ($M_w$) superimposed. The Nusfjord type-2 pseudotachylytes are shown in both the circular fault and elliptical fault cases (further elliptical aspect ratios are shown in Supplementary Fig. 5). Published seismologically determined data are included from small earthquakes at shallow depth on the San Andreas Fault[34], intraplate aftershocks from 10 to 36 km depth occurring after the 2001 Bhuj (India) earthquake[37], small earthquakes from the New Madrid seismic zone[36], and small earthquakes recorded from the Parkfield region of California[35]. Estimates from pseudotachylytes include an exhumed seismogenic fault zone in the Sierra Nevada that records seismogenic faulting at depths of 7–10 km[40] and a pseudotachylyte in lherzolites representing seismogenic faulting at >40 km[39].

constraints from microstructural and numerical modelling studies appropriate to the Nusfjord context. However, indications from models of strong inclusions within a viscously deforming matrix suggest that stresses in the inclusions can be increased by an order of magnitude[45,46] given a strength ratio >100 between the inclusion and the matrix, especially if there are additional effects resulting from interactions between strong inclusions[45], although even an isolated inclusion in an otherwise homogenous viscous matrix will invoke significant differences in stress within and around the inclusion relative to the surrounding material[47,48]. We therefore believe it feasible that the >1 GPa failure stresses required by the high seismic stress drops could be transiently reached in the internal blocks, given that the viscosity contrast between the shear zones and the dry anorthosite is much larger than that in the cited models[49].

The mechanism of rupture nucleation within relatively strong rocks in the lower crust is a new alternative to models of thermal runaway or mineral reactions that may also initiate frictional slip within otherwise viscous regimes[12,25,26] and to models of downward rupture propagation or stress pulses from shallower crustal levels[4,18]. This new mechanism is hence a somewhat

simpler explanation to account for lower crustal seismicity in continental regions that, for example, lack a major overlying fault zone, are separated by overlying seismic activity by a significant depth interval, or are thought to be anhydrous and lack evidence for eclogitisation. In the new model, the proposed stress amplification requires only the presence of a network of localised viscous shear zones[50], within a strong, dry, block-forming material that prohibits weakening and viscous creep in the internal blocks. Such conditions are found in many intraplate lower crustal granulite terranes[14,15,50,51] and this model can thus account for observations of low-magnitude present-day deep continental seismicity. This is most obviously applicable to continental settings where upper crustal seismogenic faults are not present, for example, the lower crustal seismicity of the northern Central Alpine foreland basin[29], but also along crustal-scale fault structures where deeper seismicity can be shown to be spatially and/or temporally isolated from any shallower ruptures (e.g. Baikal Rift[52], East African Rift[53]).

The new model remains compatible with the co-existence of coseismic loading of lower crustal shear zones from earthquakes nucleating in the overlying crust, as such loading can induce

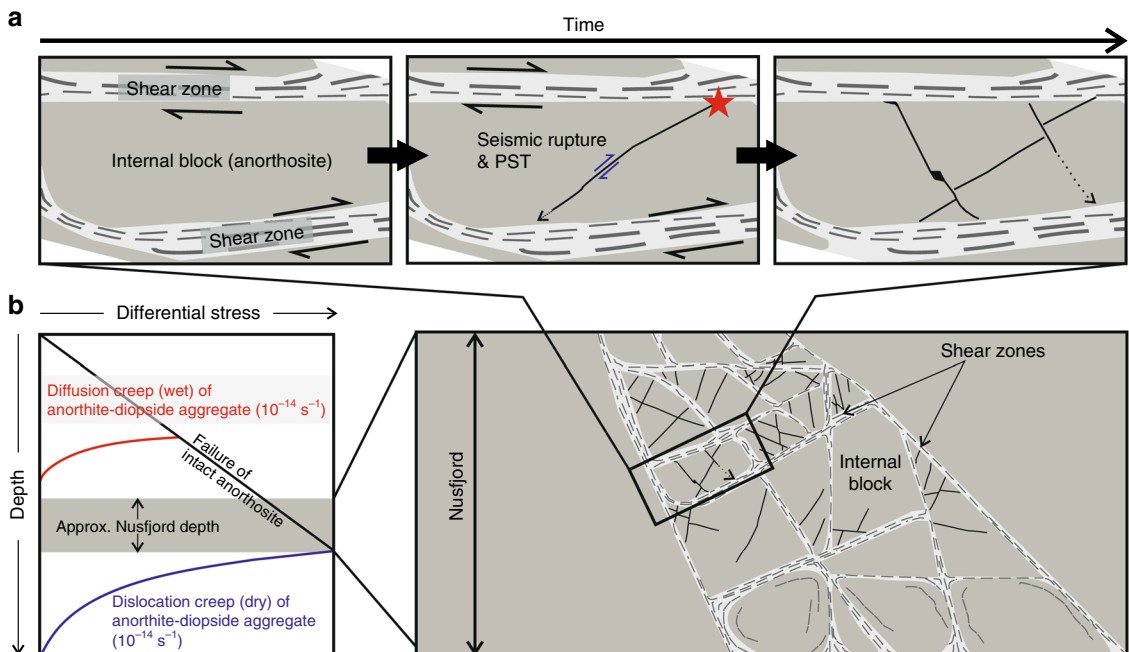

**Fig. 5 Model of pseudotachylytes in the context of the shear zone network. a** development through time of type-2 pseudotachylyte network within an internal block. Red star represents hypothetical rupture nucleation site; **b** Vertical 2D schematic section of Nusfjord shear zones showing how seismic failure of individual blocks accommodates compatibility across the shear zone network. It is possible that these shear zones could represent the roots of a crustal-scale fault system, but the geological record of this shallower crustal level is no longer preserved. Also shown are strength profiles for wet diffusion creep of anorthite-diopside aggregates[44], representing localised deformation within the shear zones[31], and for dry dislocation creep of anorthite-diopside aggregates[44] representing the hypothetical onset of viscous deformation in the anorthosite blocks.

transient increases in both the differential stress and the post-seismic strain rates across the lower crustal shear zones[49,54], resulting in an increased driving force for the internal blocks to deform. In this context, the seismicity observed in the internal blocks might represent deep aftershocks to a shallower mainshock (e.g. comparable to the deeper aftershocks of the Bhuj 2001 earthquake[6]). However, the mechanism presented here does not require upper crustal earthquakes to generate high stresses within the lower crust, only synchronous viscous deformation across a network of shear zones separating relatively high viscosity domains. Similarly, the new model does not need the ongoing or episodically triggered mineral reactions required by other in-situ models for lower crustal earthquake nucleation.

Earthquakes with high stress drops may nucleate in the dry, plagioclase-rich continental lower crust in response to locally derived stress heterogeneities. The high stresses required for failure of the strong anorthosite blocks within the shear zone network are related to coeval viscous creep across a network of highly localised shear zones mimicking the array of pre-existing tabular anisotropies, and do not need to be generated by shallower seismicity or by syn-deformation reactions. Seismic fracturing allows deformation to be kinematically sustained between adjacent shear zones and across the shear zone network as a whole.

## Methods

**Displacement and length of faults**. Several dikelets within the internal blocks act as markers for fault offset and allow the orientation of the slip vector to be calculated where two or more such markers are cut by the same fault, using the separation and offset of those markers. The displacements along pseudotachylyte faults were measured using dilational pull-aparts (Griffith et al.[40]) in order to discount any additional component of viscous displacement on the offset of markers (Fig. 3d). The measured displacements are considered the result of a single slip event, due to the lack of macroscopic reworking of the pseudotachylyte seen either in the field or from thin sections, and from the lack of fragmented or

cataclastic margins that might indicate pre-existing fault zones before melting occurred.

**Seismic parameters**. Fault length, area and displacement are input into calculations for seismic source parameters of the earthquakes which generated these pseudotachylytes. The displacement ($S$) is measured from pull-apart openings. The fault area ($A$) is derived from the fault length measured in the field, initially assuming a circular fault shape where the fault length forms the diameter. The case of an elliptical fault, where the vertical fault width extends to up to ten times the measured (horizontal) fault length, is also considered.

The moment magnitude ($M_W$) for each fault is calculated using the seismic moment ($M_0$),

$$M_0 = \mu A S \tag{1}$$

where $\mu$ is the shear modulus (38 GPa for anorthosite[55]). The moment magnitude is calculated as

$$M_W = \frac{\log M_0}{1.6} - 6.07 \tag{2}$$

The static stress drop ($\Delta\sigma$) is calculated as

$$\Delta\sigma = \frac{\mu}{C}\frac{S}{r} \tag{3}$$

where $r$ is the fault radius (for a circular fault) or semi-minor axis (for an elliptical fault) and $C$ is the geometrical coefficient calculated for transverse faults[56]. We calculate stress drops for a circular fault and an elliptical fault where the vertical extent of the fault is greater than the horizontal fault length measured in the field, with horizontal strike-slip fault movement. In this case, where $a$ is the semi-minor axis of the ellipse and is parallel to slip, and $b$ is the semi-major axis of the ellipse,

$$C = \frac{4}{3E(k) + \frac{a^2}{b^2}\frac{K(k)-E(k)}{k^2}} \tag{4}$$

where $K(k)$ and $E(k)$ are complete elliptical integrals of the first and second kind, respectively, and $k$ is defined as[40,56],

$$k = \sqrt{1 - a^2/b^2} \tag{5}$$

when the slip direction is parallel to the semi-minor axis $a$.

## Data availability

We declare that all the data used to support the conclusions of this study are accessible within the paper and its Supplementary files. The source data for Fig. 4b is displayed in Fig. 4a.

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

## Acknowledgements
This work was supported by the UK Natural Environment Research Council [grant number NE/P001548/1: The Geological Record of the Earthquake Cycle in the Lower Crust]. GP acknowledges funding from the University of Padova (BIRD175145/17: the geological record of deep earthquakes: the association pseudotachylyte-mylonite). Elisabetta Mariani and Florian Fusseis are thanked for their contribution to fieldwork and resultant discussions.

## Author contributions
L.R.C., L.M., A.F. and G.P. all undertook data collection during fieldwork, L.R.C. carried out the source parameter analysis, L.R.C., L.M., A.F. and G.P. all contributed to the interpretation and the writing and revising of the manuscript.

## Competing interests
The authors declare no competing interests.
