## [Peer Review File · Nature Communications]

Reviewers' comments:

Reviewer #1 (Remarks to the Author):

This paper presents the results of a detailed outcrop-scale study of exhumed lower crustal rocks exhibiting evidence of ductile and brittle behavior, the latter including pseudotachylyte (PST), which is widely regarded as diagnostic of formation during co-seismic slip. The specific problem being addressed is the mechanism by which deep continental crustal earthquakes, documented here by the PST and common in the earthquake record, can occur at depths typically considered to be where the crust is weak and earthquakes should not form (hypocenters at ~20-40 km), and far from plate boundaries or fault systems where shallow crustal seismicity is focused. Deep crustal earthquakes are a major conundrum for seismologists and an important problem to address as intraplate earthquakes can be highly damaging with long recurrence intervals, which tends to lull an unsuspecting, unaware public into a state of unpreparedness, increasing the cost and human toll.

The alternative mechanisms for nucleation of deep crustal earthquakes, discussed in the paper, require processes that are difficult to quantify (fluid hydration), are often the least unlikely alternative or fallback position (e.g., dynamic downward rupture on an existing fault), or simply not relevant (eclogitization in the stable non-subducting continental crust; lack of hydration of dry rocks). The authors have proposed and documented a process that is relevant to any intraplate crustal setting and is therefore a more general, widely applicable model.

With regard to the general review criteria:

-The data collected and presented are technically sound. Very detailed observations at a range of scales are presented and bolstered by additional examples in the Supplementary Materials. The critical step is in data interpretation and conversion of measured field parameters (fault length and displacement magnitudes) to parameters characteristic of upper crustal faults (moment and stress drop). The data and other observations allow for solid conclusions to be drawn. However, one minor omission relates to supporting evidence for "deep crustal" formation. The authors rely on citation of their previous petrologic work on this specific locality (Menagon et al., 2017) in support of formation of the PST by coseismic slip at 0.7-0.8 GPa and 650-750 C. That study is extremely thorough with well-supported conclusions but its importance doesn't come through in this paper. Perhaps this could be briefly expanded on or more explicitly stated how the crustal conditions were constrained.

-There is much prior work on deep crustal PST, mylonitic PST, metamorphosed PST, etc. that present convincing evidence that deep crustal earthquakes occur but without a quantitative treatment that

generates faulting/earthquake parameters that will be directly relevant to the seismological community. To my knowledge no one has recognized the significance of the geometry and style of association of brittle and ductile features as the authors have clearly done in the Nusfjord shear zone.

-In this reviewer's judgement the paper is a significant advance and likely to send other workers scurrying to find similar examples in the manner that the first descriptions of pseudotachylyte, then mylonitic pseudotachylyte, did. This work is an excellent example of how the petrologic record and careful and thorough field observations lead to practical (measurable) relevance to seismicity.

Decision: Accept with minor (suggested) editorial revisions, keyed to line numbers.

34: Delete "Recent studies highlight that,..." . You provide the citations at the end of the sentence related to the "recent studies".

41: "...25-30 km require..."

53-54: previously and then later in the paper the authors cite settings where their suggested mechanism would be relevant to nucleation of deep crustal earthquakes. East Africa and Baikal, i.e., intraplate rift settings, would be places where it is difficult to rule out dynamic downward rupture as an explanation for deep crustal seismicity and be less likely to need an alternative mechanism such as that proposed here. Also, if the new mechanism proposed here is valid one wouldn't necessarily expect to see shallow crustal earthquakes, i.e., there would be no shallow evidence of the deep crustal seismic zone. The study by Singer et al. (2014) is an excellent example for the relevance of the present study. The point here then is maybe be more explicit about where the proposed mechanism would be and wouldn't be viable.

55-58: Provide a couple more sentences at the end of the Introduction that telegraph to the reader what to expect in the "Results". Or perhaps a few sentences at the beginning of "Results". Reading through the (important) details was a bit of a slog and I stopped often to wonder where the authors were headed and why all the detail was relevant. I understand there are space constraints but for the general reader there should be SOME explanation of why the measurements that were made and descriptions provided are the KEY observations to make.

61: "...that extensively preserves..." => "...an...".

93: "...- probably..." => "interpreted to be".

95: "...picked out..." => "emphasized"

103: "...<1 m to > 10 m...": so scales could be 1 micron or 100 km? Be specific, provide lower and upper limits on scales.

147: Summarize critical observations – help the reader understand what of the previous material is important.

149-150: here is where the key evidence for "deep" crustal seismicity needs to be mentioned explicitly. Or at least move the citation #29 to the end of this sentence and move the P-T parameters in line 152 up to this first sentence.

157: "...suggests..." is not very convincing. You have made a stronger case than this.

160-163: This statement of the hypothesis should appear at the end of the Introduction.

176-178: Is it possible that there was no shallow upward continuation of the shear zone that hosts PST? I.e., such as the Alpine example of Singer et al. that doesn't appear to host shallow seismicity?

Reviewer: David P. Moecher, University of Kentucky, U.S.A.

Reviewer #2 (Remarks to the Author):

The essential claim of the paper is that large (>1GPa) stress drops occur during nucleation of earthquakes in strong, dehydrated lower crust. The latter is at the nexus over a continued debate over the mechanism by which such nucleation occurs, and as such is of broad interest to the community. The MS is well written, and arguments on the whole support conclusions. I have one

small query related to the effect of interseismic creep on the host anorthosites, as well as effect of a large static stress drop. Specifically, notwithstanding the absence of macroscopic strain in the anorthosite, is there any record of the anticipated dislocation-mediated deformation in microstructures. At the ambient temperatures, it might be expected that a 1GPa stress drop would at least influence the substructures developed around the Type 2 pst. Also, a build up in stress coeval with Type 1 shear and mylonitization would require that such shear effectively lock-up i.e. there is no path for dissipating stored energy. These two points could perhaps be clarified or expanded upon (unless I have missed something).

Review of manuscript NCOMMS-19-6025834 - Earthquake nucleation in the lower crust by local stress amplification by Campbell et al.

The authors present a very well documented field study of high-pressure high-temperature shear zones and pseudotachylytes providing insight in potential mechanisms of earthquakes affecting continental lower crust.

The three senior authors are familiar with this topic from several previous studies and the field observations and documentation are excellent and convincing. The study suggests an alternative nucleation mechanism for deep crustal earthquakes that is complementary to existing models and very well substantiated by field evidence. It provides an alternative also to other recent studies favoring depth migration of aftershocks along deep-reaching faults. The paper and in particular the well-documented observations deserve publication and certainly are of interest to a broader solid earth/geodynamics community.

Please find below a few specific comments:

L35: What is meant by dry, could that be specified? Same for impermeable, what are the estimates here?

Fig. 1: Does the red color in map and cross section indicate the anorthosite? Or is it the 'high strain zone'? What is Set 1-3 in this map?

L80 etc.: The authors suggest various rather general mechanisms leading to strain localization in the shear zones? Is detailed information on this from the field? In particular is the 'increased water content' substantiated by retrograde reactions or else? Do the shear zone rocks show evidence for dislocation creep and recrystallization? Is grain size reduction due to reaction products, recrystallization of cataclasis or all of this? The shear zones 'exploited precursor dykes and pseudotachylytes' - do most of the shear zones have brittle or magmatic precursors?

L157 etc.: What follows is largely not observations but inferences based on the observations. That part may be better suited for the discussion section.

L158 etc.: The authors suggest a mechanism that is reminiscent of the loading of rigid inclusions in a viscously deforming matrix. This is a very plausible suggestion but it does imply that strength and coupling of the deforming matrix is sufficient to load the rigid bodies up to failure. There are many analytical and modelling studies dealing with this problem. Maybe it would be possible to elaborate on this aspect a bit more and come up with some estimates bounding the failure stresses. For example, is there piezometric data from the shear zone rocks? Could the undeformed pseudotachylytes and associated fault displacements be used to estimate stresses from melting temperatures?

Fig. 4: The frictional failure line in the strength diagram does not provide an upper bound to the failure stresses, as the deformed blocks were previously intact, right? So, it is rather the fracture strength of the rock at deep crustal conditions initially limiting failure stress, I guess.

L185 etc.: The displacement-length ratio is assumed to result from a single seismic event as stated here. However, ratios of this order more typically reflect ratios of finite displacement vs length (Scholz et al., JGR 1993). So, is the observed displacement really due to a single event? Also, the authors describe ductile overprint of the pseudotachylyte margins, so there was additional slip postdating dynamic slip.

L190 etc.: The very high stress drops, much higher than estimated for aftershocks of even deep earthquakes are interesting. If I understand correctly, the fractures and shears bearing pseudotachylytes transect the entire blocks. Thus, they do not represent 'classical shear dislocation embedded in an elastic half-space' so the typical Eshelby-type source model. The problem, may be compared to the large stress drops observed in rock-mechanics failure or stick slip tests affecting a sample with finite dimensions. McGarr (GRL 2012) and McGarr et al. (BSSA, 2010) provides an attempt to correct for this effect to compare lab stress drops to those in the field which maybe also be useful here.

In summary, I enjoyed reading the paper, and I hope that my suggestions for minor revision of the manuscript are useful for the authors.

Sincerely,

Georg Dresen

We would like to thank all reviewers for their time and efforts in reviewing our work - we hope that our response has improved the manuscript and clarified our ideas to their satisfaction. Please find below our response and details of changes in the manuscript. Our response to each comment is outlined in blue italic text.

Response to reviewer 1

The authors thank David Moecher for his carefully considered review, and particularly share his hope that similar structures may indeed be recognised in other exhumed fault zones, expanding the discussion of lower crustal seismicity.

This paper presents the results of a detailed outcrop-scale study of exhumed lower crustal rocks exhibiting evidence of ductile and brittle behavior, the latter including pseudotachylyte (PST), which is widely regarded as diagnostic of formation during co-seismic slip. The specific problem being addressed is the mechanism by which deep continental crustal earthquakes, documented here by the PST and common in the earthquake record, can occur at depths typically considered to be where the crust is weak and earthquakes should not form (hypocenters at ~20-40 km), and far from plate boundaries or fault systems where shallow crustal seismicity is focused. Deep crustal earthquakes are a major conundrum for seismologists and an important problem to address as intraplate earthquakes can be highly damaging with long recurrence intervals, which tends to lull an unsuspecting, unaware public into a state of unpreparedness, increasing the cost and human toll.

The alternative mechanisms for nucleation of deep crustal earthquakes, discussed in the paper, require processes that are difficult to quantify (fluid hydration), are often the least unlikely alternative or fallback position (e.g., dynamic downward rupture on an existing fault), or simply not relevant (eclogitization in the stable non-subducting continental crust; lack of hydration of dry rocks). The authors have proposed and documented a process that is relevant to any intraplate crustal setting and is therefore a more general, widely applicable model.

With regard to the general review criteria:

-The data collected and presented are technically sound. Very detailed observations at a range of scales are presented and bolstered by additional examples in the Supplementary Materials. The critical step is in data interpretation and conversion of measured field parameters (fault length and displacement magnitudes) to parameters characteristic of upper crustal faults (moment and stress drop). The data and other observations allow for solid conclusions to be drawn. However, one minor omission relates to supporting evidence for “deep crustal” formation. The authors rely on citation of their previous petrologic work on this specific locality (Menegon et al., 2017) in support of formation of the PST by coseismic slip at 0.7-0.8 GPa and 650-750 C. That study is extremely thorough with well-supported conclusions but its importance doesn’t come through in this paper. Perhaps this could be briefly expanded on or more explicitly stated how the crustal conditions were constrained.

With reference to the general point above that the evidence for deep crustal formation could be emphasised further, we have restructured the paragraph at L166 (‘Evidence for earthquake nucleation within the lower crust’) to emphasise the important observations that lead us to this

conclusion, and have also detailed the methods used in Menegon et al., 2017 that resulted in the pressure and temperature constraints (L169).

-There is much prior work on deep crustal PST, mylonitic PST, metamorphosed PST, etc. that present convincing evidence that deep crustal earthquakes occur but without a quantitative treatment that generates faulting/earthquake parameters that will be directly relevant to the seismological community. To my knowledge no one has recognized the significance of the geometry and style of association of brittle and ductile features as the authors have clearly done in the Nufsjord shear zone.

-In this reviewer's judgement the paper is a significant advance and likely to send other workers scurrying to find similar examples in the manner that the first descriptions of pseudotachylyte, then mylonitic pseudotachylyte, did. This work is an excellent example of how the petrologic record and careful and thorough field observations lead to practical (measurable) relevance to seismicity.

Decision: Accept with minor (suggested) editorial revisions, keyed to line numbers.

All other changes have been outlined after the relevant comment below:

34: Delete "Recent studies highlight that,...". You provide the citations at the end of the sentence related to the "recent studies".

This has been removed as suggested.

41: "...25-30 km require..."

This has been altered as suggested.

53-54: previously and then later in the paper the authors cite settings where their suggested mechanism would be relevant to nucleation of deep crustal earthquakes. East Africa and Baikal, i.e., intraplate rift settings, would be places where it is difficult to rule out dynamic downward rupture as an explanation for deep crustal seismicity and be less likely to need an alternative mechanism such as that proposed here. Also, if the new mechanism proposed here is valid one wouldn't necessarily expect to see shallow crustal earthquakes, i.e., there would be no shallow evidence of the deep crustal seismic zone. The study by Singer et al. (2014) is an excellent example for the relevance of the present study. The point here then is maybe be more explicit about where the proposed mechanism would be and wouldn't be viable.

We agree that the Singer et al. (2014) study of deep seismicity in the bend of the down-going plate of the Alpine foreland is a highly appropriate example of modern-day seismicity where our model can explain the deep earthquake nucleation, without any ambiguity on whether shallower crustal faulting might be an influence. However, we suggest that, despite the different structural context, our model is not inviable for intraplate rift settings; although there are shallow crustal earthquakes in the Baikal and East African rifts, they are too small in magnitude (at least in the recorded era) and in

some regions separated by too great a distance (either up-fault or along strike, or indeed temporally) from the deeper seismicity to viably influence the stress field at the level of the deeper quakes (e.g. Nyblade & Langston, 1995). We wish to keep these references to frame our argument that our model is viable whether a shallower fault zone is present or not. However, we have tried to emphasise that the Singer et al. (2014) study is of particular relevance by explicitly referencing the study at this point (L56) and by outlining the above argument in the discussion section around L273.

55-58: Provide a couple more sentences at the end of the Introduction that telegraph to the reader what to expect in the “Results”. Or perhaps a few sentences at the beginning of “Results”. Reading through the (important) details was a bit of a slog and I stopped often to wonder where the authors were headed and why all the detail was relevant. I understand there are space constraints but for the general reader there should be SOME explanation of why the measurements that were made and descriptions provided are the KEY observations to make.

We have added an outline of the results section here at the end of the introduction, L57-62.

61: “...that extensively preserves...” => “..an...”.

93: “...- probably...” => “interpreted to be”.

95: “...picked out...” => “emphasized”

The three changes above have been implemented.

103: “...<1 m to > 10 m...”: so scales could be 1 micron or 100 km? Be specific, provide lower and upper limits on scales.

We have changed this to ‘1m to 15 m’ to constrain the length range of the examples presented here.

147: Summarize critical observations – help the reader understand what of the previous material is important.

149-150: here is where the key evidence for “deep” crustal seismicity needs to be mentioned explicitly. Or at least move the citation #29 to the end of this sentence and move the P-T parameters in line 152 up to this first sentence.

This paragraph has been restructured in order to outline the evidence better. We state the field observation that the pseudotachylytes are coeval with the mylonitisation on the shear zones, detail the methods in citation 31 (previously #29) that allow us to constrain that mylonitisation to the lower crust, and then support this with the observation that there is no retrogression observed in the pseudotachylyte veins or in the damage zones to those faults. We hope that this is now clearer.

157: “...suggests...” is not very convincing. You have made a stronger case than this.

‘suggest’ is replaced by ‘reveals’

160-163: This statement of the hypothesis should appear at the end of the Introduction.

We have added a similar statement at the end of the introduction, L62, but feel it is also important to emphasise here in the discussion too.

176-178: Is it possible that there was no shallow upward continuation of the shear zone that hosts PST? I.e., such as the Alpine example of Singer et al. that doesn't appear to host shallow seismicity?

We certainly are open to this being the case – actually there is no evidence preserved in Nusfjord as to whether or not any shallower structures were present, and we would like to emphasise that a feature of the model is that it is applicable in either scenario. This is discussed in L266 onwards, including a reference to Singer et al., (2014).

Reviewer: David P. Moecher, University of Kentucky, U.S.A.

Response to reviewer 2:

The essential claim of the paper is that large (>1GPa) stress drops occur during nucleation of earthquakes in strong, dehydrated lower crust. The latter is at the nexus over a continued debate over the mechanism by which such nucleation occurs, and as such is of broad interest to the community. The MS is well written, and arguments on the whole support conclusions. I have one small query related to the effect of interseismic creep on the host anorthosites, as well as effect of a large static stress drop. Specifically, notwithstanding the absence of macroscopic strain in the anorthosite, is there any record of the anticipated dislocation-mediated deformation in microstructures. At the ambient temperatures, it might be expected that a 1GPa stress drop would at least influence the substructures developed around the Type 2 pst. Also, a build up in stress coeval with Type 1 shear and mylonitization would require that such shear effectively lock-up i.e. there is no path for dissipating stored energy. These two points could perhaps be clarified or expanded upon (unless I have missed something).

The authors thank reviewer 2 for some thought-provoking comments, these two points are indeed interesting to consider and have stimulated us to examine some further samples, images of which are now included in the supplementary information.

In regards to the first point (whether there is any record of dislocation-mediated deformation surrounding the type-2 pseudotachylytes), there is some limited evidence of dislocation glide in the anorthosite, especially in the margins of the pseudotachylyte-bearing faults. The most obvious deformation in the anorthosite, however, is microfracturing, which does appear in the field to be concentrated within the potential damage zone of both type-1 and type-2 pseudotachylytes (Fig. 3f, also see new Supp. Fig. 2a). We do see some microstructures in the plagioclase and in the pyroxene that are consistent with low temperature plasticity and dislocation glide, for example, some patchy or undulose distortion in crystal lattice between (and cut by) the microfractures (new Supp. Fig. 2a). These microstructures, as well as the lack of evidence for dislocation creep in plagioclase here at 700°C (Supp. Fig. 2b), are consistent with relatively high stresses (and/or strain rates) – it is, however, difficult to say at this stage with the limited microstructural component of this contribution whether this high stress deformation occurred as a response to the cycles of stress amplification that we discuss here, or if it could also be introduced by coseismic rupture effects and damage, or otherwise. In fact, we have other work in preparation which will explore exactly this. We have added a few sentences on these microstructural observations in the anorthosite at L 97.

With regard to the second point, we do not envisage that the bounding shear zones necessarily lock up completely, as we have no evidence for any widespread hardening within the mylonites in these shear zones. We do invoke stress cycles which increase towards the failure strength before being released with seismic failure, as in the model of Passchier (1982) for the cyclical production of pseudotachylyte in association with mylonites in the Saint Barthélemy Massif (see particularly his Fig. 11) – his model does invoke hardening of the host rocks and coupled locking of the shear zones prior to seismic failure, as the ruptures there occur within the mylonitic shear zones. However, in the case of our internal block ruptures, we suggest that the equivalent ‘hardening’ in the Nusfjord case is purely a geometrical effect of the shear zone/internal block network which locally amplifies the stresses at certain points within the internal blocks (leading to rupture nucleation). We do not mean

to imply any work hardening mechanisms or a complete stoppage of deformation within any bounding shear zone (as in this case the drivers for the block to fail between the locked shear zones would be removed). We do not invoke any rheological hardening of the blocks either, because in our view the blocks are strong in the first place (because they are dry). Due to their high creep strength, they are unable to accommodate internal viscous deformation necessary for the displacement along the bounding shear zones, and instead elastic strain builds up. This causes stress amplifications that locally lead to seismic failure in the anorthosite blocks. We have sought to explain our ideas in greater detail within the manuscript at L249 onwards.

Response to reviewer 3

The authors thank Georg Dresen for his review, which has resulted in some important additional details and quantitative values being added into the manuscript, as well as some interesting ideas on fault scaling and context.

The authors present a very well documented field study of high-pressure high-temperature shear zones and pseudotachylytes providing insight in potential mechanisms of earthquakes affecting continental lowercrust.

The three senior authors are familiar with this topic from several previous studies and the field observations and documentation are excellent and convincing. The study suggests an alternative nucleation mechanism for deep crustal earthquakes that is complementary to existing models and very well substantiated by field evidence. It provides an alternative also to other recent studies favoring depth migration of aftershocks along deep-reaching faults. The paper and in particular the well- documented observations deserve publication and certainly are of interest to a broader solid earth/geodynamics community.

Please find below a few specific comments:

L35: What is meant by dry, could that be specified? Same for impermeable, what are the estimates here?

We have reconsidered and removed 'impermeable', firstly because the shear zones/type -1 pseudotachylytes could provide a fluid pathway if fluid was available (Menegon et al., 2017) and secondly because if there is no provision for fluid, a dry rock will remain dry no matter how permeable or not the rock is.

There is not a huge amount of literature values available to quantify exactly what qualifies as 'dry' for natural examples of lower crustal rocks. One granulite assemblage entirely consisting of anhydrous minerals is calculated to have 0.015–0.095 wt. % H₂O (Xia et al. 2006). The Nusfjord anorthosites are calculated to have 0.05 wt. % H₂O in the host rock, rising to 0.4 wt. % within the type-1 mylonitised pseudotachylytes (Menegon et al. 2017). We have added these values for the Nusfjord pseudotachylytes in the results section characterising the shear zone network (L96), but prefer in this introductory section to keep the concept quite general, however we do now note the work of Xia et al. (2006) in L39.

Fig. 1: Does the red color in map and cross section indicate the anorthosite? Or is it the 'high strain zone'? What is Set 1-3 in thismap?

The red shading does indicate the high strain zone, this has been clarified in the figure caption. We have added labels on the map to distinguish the different shear zone sets.

L80 etc.: The authors suggest various rather general mechanisms leading to strain localization in the shear zones? Is detailed information on this from the field? In particular is the 'increased

water content' substantiated by retrograde reactions or else? Do the shear zone rocks show evidence for dislocation creep and recrystallization? Is grain size reduction due to reaction products, recrystallization of cataclasis or all of this? The shear zones 'exploited precursor dykes and pseudotachylytes' - do most of the shear zones have brittle or magmatic precursors?

We have added more detail in this section (L92 onwards) to clarify that these are observed differences between the precursor structures and the surrounding anorthosite in Nusfjord that are interpreted to have facilitated the strong strain localization. Many of these points are presented in Menegon et al. (2017) and we have cited this more frequently to make this clear.

The 10 x increase in water content in the mylonitised pseudotachylytes relative to the host anorthosite was calculated via thermodynamic modelling of fluid contents required for the mylonitised pseudotachylyte assemblage (Menegon et al., 2017) and is interpreted to be a consequence of coseismic fluid redistribution (Menegon et al., 2017, Jamtveit et al., 2018).

Within the mylonitised pseudotachylytes, the dominant mechanism was grain size sensitive creep of the fine-grained pseudotachylyte assemblage (Menegon et al., 2017). Within the shear zones but outside the individual type-1 pseudotachylyte veins (see e.g. Campbell & Menegon 2019 Figs. 2-4) the plagioclase (and any pyroxene) is predominantly fractured but quartz aggregate domains, where present, show evidence for dislocation creep and dynamic recrystallisation, particularly near the pseudotachylyte veins (Campbell & Menegon, 2019).

The reduced grain size relative to much of the anorthosite is a primary feature of the intrusions and the pseudotachylytes which act as the precursor structures. All of the shear zones that we have seen appear to have had a brittle and/or magmatic precursor – we suggest that the low strain example in Fig. 2a illustrates the pre-mylonitised nature of these precursors, with pseudotachylytes also exploiting mechanical anisotropy introduced by dykes.

L157 etc.: What follows is largely not observations but inferences based on the observations. That part may be better suited for the discussion section.

We have moved this section to the start of the discussion.

L158 etc.: The authors suggest a mechanism that is reminiscent of the loading of rigid inclusions in a viscously deforming matrix. This is a very plausible suggestion but it does imply that strength and coupling of the deforming matrix is sufficient to load the rigid bodies up to failure. There are many analytical and modelling studies dealing with this problem. Maybe it would be possible to elaborate on this aspect a bit more and come up with some estimates bounding the failure stresses. For example, is there piezometric data from the shear zone rocks? Could the undeformed pseudotachylytes and associated fault displacements be used to estimate stresses from melting temperatures?

We agree that our ideas presented here on block deformation within shear zones are

comparable with modelling of rigid inclusions within viscous matrices and have extended our citations of some related studies to support our argument for quite localized stress amplification within each internal block, related to the bounding shear zone geometry (e.g. Mancktelow, 2002), as well as to identify controls on the magnitude of stress amplification in weak matrix-rigid clast scenarios (Beall et al., 2019, Zhao & Ji, 1997). We are cautious of implying that the strength of the bounding shear zone is completely coupled to that of the internal block (see also our reply to a related comment by reviewer 2) as we see no evidence in the mylonites for strain hardening. We outline in more detail the mechanisms of stress amplification and cycles within the internal block from L246 onwards.

We have recently published piezometric calculations of transient high stress episodes in the bounding shear zones reaching 100 MPa (Campbell & Menegon, 2019) but these are not related to the stress amplification cycles that we postulate here, being instead the preservation of highly localised high strain rate deformation during postseismic creep after the type-1 pseudotachylyte generation event (i.e. these stresses are lower than the differential stresses required for frictional and seismic failure of intact anorthosites at lower crustal depths, which are on the order of several hundreds of MPa to > 1 GPa, depending on the tectonic setting).

For the failure stresses, we remain happy to use the maximum static stress drop as an indication of the failure strength of the anorthosite, as it is in line with theoretical failure strength of intact anorthosite at these depths. It is interesting to look at displacements and pseudotachylyte melt temperatures and volumes as an indicator of shear stress, but because the shear stress falls dynamically during coseismic slip, and the ongoing melt production reflects this, it is difficult to use these observations to constrain the initial failure shear stress (as far as we are aware) - instead, typically such observations are used to calculate an average of the dynamic shear stress distribution (e.g. Di Toro et al., 2005). We do have some data in this context based on measurements of internal block pseudotachylyte displacement and thickness, but as they are not necessarily the same set of faults used in the source parameter calculations here (Fig. 4) we do not think that they add enough in this contribution to include them.

Fig. 4: The frictional failure line in the strength diagram does not provide an upper bound to the failure stresses, as the deformed blocks were previously intact, right? So, it is rather the fracture strength of the rock at deep crustal conditions initially limiting failure stress, I guess.

We agree with this observation, to emphasize this we have changed this (now Fig. 5b) to 'frictional failure of intact anorthosite'.

L185 etc.: The displacement-length ratio is assumed to result from a single seismic event as stated here. However, ratios of this order more typically reflect ratios of finite displacement vs length (Scholz et al., JGR 1993). So, is the observed displacement really due to a single event? Also, the authors describe ductile overprint of the pseudotachylyte margins, so there was

additional slip postdating dynamic slip.

The applicability of scaling relationships to small faults and to the lower crust is an interesting and topical discussion point, which we believe currently suffers from a dearth of data and hence hope that the observations in this manuscript will provide an important contribution to. Firstly, we are confident that each type-2 pseudotachylyte fault presented here represents a single coseismic slip event. There is no evidence for a brittle overprint of these pseudotachylyte veins, nor is there evidence that these pseudotachylytes overprinted a pre-existing fault, in which case we might expect to have seen cataclasite preserved in survivor clasts or perhaps in the vein margins.

The pseudotachylyte vein shown in Fig. 3d shows a solid state viscous overprint localized to a 1 mm wide domain in the margin of this vein. However, this viscous contribution to displacement is a maximum of 1 mm, based on strain analysis of deflected markers in the deformed spherulite layer. This vein was not incorporated into the dataset of fault length and displacements used to calculate source parameters. Given typical displacements of 1-30 cm, though, this viscous overprint is negligible. We do not expect that the pseudotachylytes that we measured for length and displacement would have any more significant viscous overprint than this, because we checked in the field for signs of such overprints and recognized none. Other thin sections from type-2 pseudotachylytes do not show any solid-state viscous overprints – we have added a micrograph of another type-2 sample in Supp. Fig. 2a and some additional text in the manuscript around L161-165.

Finally, our observations of large displacement to length ratios are consistent with records of other lower crustal pseudotachylytes in the Bergen Arcs (e.g. Petley-Ragan et al., 2019). It may be that earthquakes at depth do not conform to typical scaling relationships based on upper crustal faults. Our faults, however, do differ from many others (whether upper or lower crustal) in that their length and area are constrained by the size and geometry of the internal block, i.e. the maximum fault area is fixed prior to rupture. Hence, we do not push for conformity to any such scaling relationships.

L190 etc.: The very high stress drops, much higher than estimated for aftershocks of even deep earthquakes are interesting. If I understand correctly, the fractures and shears bearing pseudotachylytes transect the entire blocks. Thus, they do not represent ‘classical shear dislocation embedded in an elastic half-space’ so the typical Eshelby-type source model. The problem, may be compared to the large stress drops observed in rock-mechanics failure or stick slip tests affecting a sample with finite dimensions. McGarr (GRL 2012) and McGarr et al. (BSSA, 2010) provides an attempt to correct for this effect to compare lab stress drops to those in the field which maybe also be useful here.

We agree that these faults are probably not appropriately represented by the Eshelby source model due to the fixed maximum area of the fault (many do cut through the entire block, although some, particularly those seen in the locality Fig. 3b, do not always show the pseudotachylyte vein continuing across the entire block and it is not always clear whether a barren portion of the same fault has propagated further). We use the Eshelby-type stress drop as a well-understood

comparison to seismological stress drop values, however we also now note that stress drops may be large because the rupture zone is limited by the size of the block, analogous to laboratory experiments (L218). We are intrigued by the possibility to correct for this effect, but find that the correction requires estimating several parameters (such as the stiffness of the rocks surrounding the fault, static and dynamic frictional properties, and fault zone normal stress) which we do not know well. We therefore prefer to use the Eshelby source with the caveat of limited rupture dimension.

References cited in response

- Beall, A., Fagereng, Å. & Ellis, S. Strength of Strained Two-Phase Mixtures: Application to Rapid Creep and Stress Amplification in Subduction Zone Mélange. *Geophysical Research Letters* **46**, 169–178 (2019).
- Campbell, L. R. & Menegon, L. Transient High Strain Rate During Localized Viscous Creep in the Dry Lower Continental Crust (Lofoten, Norway). *J. Geophys. Res. Solid Earth* **n/a**, (2019)
- Di Toro, G., Pennacchioni, G. & Teza, G. Can pseudotachylytes be used to infer earthquake source parameters? An example of limitations in the study of exhumed faults. *Tectonophysics* **402**, 3–20 (2005).
- Jamtveit, B., Ben-Zion, Y., Renard, F. & Austrheim, H. Earthquake-induced transformation of the lower crust. *Nature* **556**, 487–491 (2018).
- Mancktelow, N. S. Finite-element modelling of shear zone development in viscoelastic materials and its implications for localisation of partial melting. *J. Struct. Geol.* **24**, 1045–1053 (2002).
- Menegon, L., Pennacchioni, G., Malaspina, N., Harris, K. & Wood, E. Earthquakes as Precursors of Ductile Shear Zones in the Dry and Strong Lower Crust. *Geochemistry, Geophys. Geosystems* **18**, 4356–4374 (2017).
- Passchier, C. W. Pseudotachylyte and the development of ultramylonite bands in the Saint-Barthélemy Massif, French Pyrenees. *J. Struct. Geol.* **4**, 69–79 (1982).
- Petley-Ragan, A. *et al.* Dynamic earthquake rupture in the lower crust. *Sci. Adv.* **5**, eaaw0913 (2019).
- Xia, Q.-K., Yang, X.-Z., Deloule, E., Sheng, Y.-M. & Hao, Y.-T. Water in the lower crustal granulite xenoliths from Nushan, eastern China. *J. Geophys. Res. Solid Earth* **111**, (2006). Zhao, P. & Ji, S. Refinements of shear-lag model and its applications. *Tectonophysics* **279**, 37–53 (1997).
- Zhao, P. & Ji, S. Refinements of shear-lag model and its applications. *Tectonophysics* **279**, 37–53 (1997).

REVIEWERS' COMMENTS:

Reviewer #1 (Remarks to the Author):

I reviewed this manuscript upon its initial submission to Nature Comms. I read the authors' rebuttal/responses to my initial review comments and am satisfied that they have adequately addressed my suggestions. The only item I could find remaining to be fixed is in Figure 5a: explain the red star.

Reviewer #2 (Remarks to the Author):

The submission by Campbell et al. addresses an important and controversial phenomenon observed throughout the middle and deep crust; that is, given both seismological and geological evidence for nucleation of earthquakes in these regimes, what is nature of seismic nucleation? At the core of the issue is the pressure dependence of standard fracture/friction relationships that predict failure stresses much greater than either anticipated crustal stresses or observed earthquake stress drops. To explain the occurrence of seismic fault melts (pseudotachylite) at depth, various scenarios involving propagation of faults from the upper crustal seismogenic zone into the ductile regime have been suggested. An alternative proposal to brittle rupture has been thermo-mechanical runaway wherein ductile shear and thermal feedback lead to rapid slip and melting consisting with brittle fault melts.

A third possibility is argued by the authors and forms the essential statement of their contribution - rupture is nucleated within the lower crust in response to localized stress concentrations in strong materials generated at the contacts with high-strain-rate ductile (viscous) flow. The mechanism itself is one suggested at various times throughout the history of pseudotachylite research (e.g. Sibson 1979, *J. Struc. Geol.*), though without the detailed observations presented here.

Notwithstanding the excellent field observations and interpretations, there is one critical topic that further comment, or at least an indication of need for further research would strengthen the case made by the authors. Specifically, this is whether stress amplification of the required magnitude can be achieved. The latter is a long-standing area of study beginning with Eshelby (1959) for solely elastic concentrations and Bilby et al. (1975) for visco-elastic behaviour, and is characterized by recent determinations by Zhong, Dabrowski and Jamtveit (2019), *Geophys. J. Int.* A quick (and non-systematic) review of modeled amplifications show typical stress concentrations of 20-30% for a 10-fold contrast in shear modulus. Under certain conditions, gains of 200-300% might be possible, but for observed stress drops, this still puts the stress amplification below the GPa level. Although the stress amplification seems feasible, it would be more satisfying to know it is possible.

With the above proviso, I have an overall positive view of the submission, and it is certainly timely in its content (e.g. Royal Society Mtg, Feb. 2020). Specific comments on the text are made below. All comments and suggestions are intended in a collegial manner and I hope they are useful to the authors.

I. 50 replace “hypothesized” with “proposed”?

I. 63 “strain compatibility” – perhaps too much emphasis on strain compatibility – i.e. don’t tell the rocks how they must deform

I. 124 Fig. 2 caption - change “flattening” to something like “alignment of deformed, elongate clasts...” – this avoids inferring a deformation path that is unknown and exchanges genetic for descriptive label

I. 133 “crystallized (inferred coseismic) melts” – brackets are an interpretation, even if true whereas melt is demonstrable

I. 201 – “..deform coherently” – models have specific and not necessarily relevant boundary conditions whereas natural deformations do not require mathematical tractability – to me this implies something that is not rigorously established for the rock mass; i.e. the system deforms as it wishes, and incoherence and incompatibility is actually necessary for development of significant stress deviations.

I. 242, here or somewhere in the text, probably should refer to Sibson (1979) – although the crustal depths are different, reference was made to stress concentrations around mafic bodies nucleating rupture

I. 251 – “frictional failure strength of intact anorthosite” is a bit ambiguous (to me) – do you mean “fracture of intact anorthosite and subsequent frictional sliding”

I. 253 – “elastic strain accumulation” to “elastic energy accumulation” ?– from text, both stress and strain are increasing

l. 255 – at end of sentence “ that if high enough could lead to seismic failure”

l.286 – given the arguments for amplification of stress, would not the argument be that “strain incompatibility of the deforming system was accommodated by transient seismic failure” – the overall deformation is distinctly heterogeneous

l. 292 – 303 – this is where some discussion or reference to elastic and viscous contrasts giving stress variations could be included (see general comment)

l. 327 – replace “instigate” with induce”?

l. 336 – stress concentrations – again, can this occur a là Eshelby, entrained deformation, etc

l. 341 – “allows deformation to be kinematically suststained”?

We would like to thank both reviewers and the editorial team for their time and efforts in thoughtfully considering our manuscript, and we hope that our response here and corresponding revisions have improved the work to the satisfaction of all. Please find outlined here our response to the reviewers comments, with details of the relevant changes to the manuscript. Our response to each comment is highlighted in blue text.

REVIEWERS' COMMENTS:

Reviewer #1 (Remarks to the Author):

I reviewed this manuscript upon its initial submission to Nature Comms. I read the authors' rebuttal/responses to my initial review comments and am satisfied that they have adequately addressed my suggestions. The only item I could find remaining to be fixed is in Figure 5a: explain the red star.

We thank reviewer 1 for reconsidering the manuscript and are pleased to have addressed all previous concerns. We have added to the figure caption of Fig. 5a 'Red star represents hypothetical rupture nucleation site'.

Reviewer #2 (Remarks to the Author):

The submission by Campbell et al. addresses an important and controversial phenomenon observed throughout the middle and deep crust; that is, given both seismological and geological evidence for nucleation of earthquakes in these regimes, what is nature of seismic nucleation? At the core of the issue is the pressure dependence of standard fracture/friction relationships that predict failure stresses much greater than either anticipated crustal stresses or observed earthquake stress drops. To explain the occurrence of seismic fault melts (pseudotachylyte) at depth, various scenarios involving propagation of faults from the upper crustal seismogenic zone into the ductile regime have been suggested. An alternative proposal to brittle rupture has been thermo-mechanical runaway wherein ductile shear and thermal feedback lead to rapid slip and melting consisting with brittle fault melts.

A third possibility is argued by the authors and forms the essential statement of their contribution - rupture is nucleated within the lower crust in response to localized stress concentrations in strong materials generated at the contacts with high-strain-rate ductile (viscous) flow. The mechanism itself is one suggested at various times throughout the history of pseudotachylyte research (e.g. Sibson 1979, J. Struc. Geol.), though without the detailed observations presented here.

Notwithstanding the excellent field observations and interpretations, there is one critical topic that further comment, or at least an indication of need for further research would strengthen the case made by the authors. Specifically, this is whether stress amplification of the required magnitude can be achieved. The latter is a long-standing area of study beginning with Eshelby (1959) for solely elastic concentrations and Bilby et al. (1975) for visco-elastic behaviour, and is characterized by recent determinations by Zhong, Dabrowski and Jamtveit (2019), Geophys. J. Int. A quick (and non-systematic) review of modeled amplifications show typical stress concentrations of 20-30% for a 10-fold contrast in shear modulus. Under certain conditions, gains of 200-300% might be possible, but for observed stress drops, this still puts the stress amplification below the GPa level. Although the stress amplification seems feasible, it would be more satisfying to know it is

possible.

With the above proviso, I have an overall positive view of the submission, and it is certainly timely in its content (e.g. Royal Society Mtg, Feb. 2020). Specific comments on the text are made below. All comments and suggestions are intended in a collegial manner and I hope they are useful to the authors.

We thank reviewer 2 for their considered comments and particularly for prompting us to refine our discussion of the level of stress amplification that we propose. To address this major comment, we have considered the suggested literature alongside some of the studies we cite already in order to more explicitly evaluate feasible stress increases. In terms of future work, we are already in the process of evaluating whether stresses in these internal blocks can be constrained microstructurally, both via characterisation of the deformation mechanisms exhibited in the various anorthosite mineral phases and via HR-EBSD analysis of residual stresses. To expand this further, we also plan to numerically model the geometry and rheologies observed in the field observations presented here in order to systematically test the spatial heterogeneity of stress changes. Both of these strands of investigation are significant blocks of work in themselves, and we intend to submit them as separate contributions for publication. We include a note in the manuscript to this effect at L207.

We take the reviewers suggestion to add extra references to help quantify the maximum stress amplifications possible. We note, however, that several models of strong inclusions in weaker viscously deforming matrix tend to consider a single isolated inclusion within a relatively large volume of viscous matrix. Whilst still invoking stress and/or strain concentrations in the inclusion, these models may not be as appropriate to the Nufjurd shear zones as models which include several, potentially interacting, strong inclusions and (ideally) a relatively low volume of matrix. For example, we add the suggested citation to Eshelby (1957) but note that the stress field within his modelled inclusion is homogeneous, due to the homogenous and infinite nature of the surrounding medium. Despite this, analogue and numerical modelling of single strong inclusions in a weaker matrix has observed a 40-fold increase in stresses inside the inclusion when the strength ratio (elastic moduli) is 0.02 between matrix and inclusion. For smaller ratios of 0.1, a 7-fold stress increase in the inclusion can still be gained (Zhao and Ji, 1997). This study also showed that increasing aspect ratios and volume fractions of the strong phase are also significant drivers of increased stress concentrations, as well as the strength contrast.

Another existing study relevant to this discussion (which is already cited elsewhere in the manuscript) is Beall et al. (2019). This study shows that shear stresses in strong inclusions can be amplified up to 14 times that of the applied stress if there is also some force-chain effect between inclusions (even if the chain is not extensive), using a viscosity ratio between matrix and inclusion of 0.001. At lower viscosity contrasts of 0.01, the stress concentration was still up to 5.5 times higher in the inclusion. These viscosity contrasts are smaller than we predict for Nufjurd, where the anorthosite is likely to exhibit very high viscosity of $\sim 10^{24}$ Pa. s., whilst the shear zones may transiently display viscosities as low as 10^{16} Pa. s. (Campbell & Menegon, 2019). The volume fractions of weak shear zone to strong inclusion that we observe in Nufjurd are not commonly approximated in existing models, which leads us to believe that further modelling work could be very useful in quantifying these stresses. This discussion has been included in the manuscript at L 206-219

I. 50 replace “hypothesized” with “proposed”?

We have changed this as suggested (L45)

I. 63 “strain compatibility” – perhaps too much emphasis on strain compatibility – i.e. don’t tell the rocks how they must deform

We have changed this to ‘*We interpret this mechanism of seismicity to be a mechanical response to strain incompatibility across the shear zone network during localised viscous shear*’

I. 124 Fig. 2 caption - change “flattening” to something like “alignment of deformed, elongate clasts...” – this avoids inferring a deformation path that is unknown and exchanges generic for descriptive label

We have changed this as suggested (L455)

I. 133 “crystallized (inferred coseismic) melts” – brackets are an interpretation, even if true whereas melt is demonstrable

We have adjusted to ‘undeformed and unaltered from their origin as crystallized melts, inferred to be coseismic’ (L101).

I. 201 – “..deform coherently” – models have specific and not necessarily relevant boundary conditions whereas natural deformations do not require mathematical tractability – to me this implies something that is not rigorously established for the rock mass; i.e. the system deforms as it wishes, and incoherence and incompatibility is actually necessary for development of significant stress deviations.

We accept this point and have removed ‘in order to allow the whole high and low strain system to deform coherently’.

I. 242, here or somewhere in the text, probably should refer to Sibson (1979) – although the crustal depths are different, reference was made to stress concentrations around mafic bodies nucleating rupture

This is an intriguing comparison, although we note one major difference in the model of Sibson (1980): there, the strain and strain rate are enhanced in the viscously deforming material immediately adjacent to the strong lens, but are not considered within the lens. We do not wish to suggest that the implication in Nusfjord would be that the earthquakes nucleated in the shear zones and propagated into the blocks, which we do not see any evidence for. Sibson (1980) does not explicitly consider the stress inside the block, but studies that also evaluate stress concentrations around the block (Zhao & Ji, 1997) observe even greater stress concentration inside the block. In the current manuscript we now cite Sibson (1980) at L 184 and L 216 (ref. 48) as part of the extended discussion on literature treatment of strong inclusions.

I. 251 – “frictional failure strength of intact anorthosite” is a bit ambiguous (to me) – do you mean “fracture of intact anorthosite and subsequent frictional sliding”

We accept that this statement is a bit ambiguous, we have clarified this as ‘consistent with both the high strength of anorthite reported from experimental studies⁴³ and the stresses required for failure of intact anorthosite and subsequent frictional sliding at high lower crustal confining pressures.’ (L172).

I. 253 – “elastic strain accumulation” to “elastic energy accumulation” ?– from text, both stress and strain are increasing

This is true, we have changed this as suggested (L175).

I. 255 – at end of sentence “ that if high enough could lead to seismic failure”

Actually, we are reluctant to enact this change that we believe the reviewer is suggesting for the sentence ‘High viscous strength in the anorthosite blocks would enable seismic failure through elastic energy accumulation, because the dry, coarse-grained plagioclase could not flow viscously – even at geological strain rates - without a reduction in grain size, changed mineralogy, or fluid influx^{43,44}’ because the processes mentioned at the end are called upon as potential facilitators of viscous deformation, and the absence of viscous deformation (in the internal blocks) is allowing high stresses and strains to build. Hence the reviewer’s suggestion would not help convey this meaning.

I.286 – given the arguments for amplification of stress, would not the argument be that “strain incompatibility of the deforming system was accommodated by transient seismic failure” – the overall deformation is distinctly heterogeneous

We agree that the system of deformation is completely heterogeneous and, in our suggested model, is for the majority of its active deformation characterised by varying levels of strain incompatibility. Our meaning with this sentence was to argue that over long time scales, the consequence of episodic seismicity would be to approximate long term strain compatibility, allowing continuous viscous deformation, even whilst at any point in time it would be found to be in a state of incompatibility. We feel here that the reviewer’s suggestion is an improvement on our phrasing and have replaced the suggested sentence. We further clarify our position detailed in this response with the addition of the sentence ‘Over long timescales, the effect of episodic seismic activity was to approximate strain compatibility across the shear zones, at least enough to facilitate ongoing viscous deformation’. (L187)

I. 292 – 303 – this is where some discussion or reference to elastic and viscous contrasts giving stress variations could be included (see general comment)

We include further discussion to this effect at 207-219 - please see our response to the general comments for more details

I. 327 – replace “instigate” with induce”?

We have made this change as suggested (L236).

I. 336 – stress concentrations – again, can this occur a la Eshelby, entrained deformation, etc

We include all discussion on this at L207-219

I. 341 – “allows deformation to be kinematically sustained”?

We have made this change as suggested (L250).